# Sample Efficiency Matters: Training Multimodal Conversational Recommendation Systems in a Small Data Setting

## ABSTRACT

With the increasing prevalence of virtual assistants, multimodal conversational recommendation systems (multimodal CRS) becomes essential for boosting customer engagement, improving conversion rates, and enhancing user satisfaction. Yet conversational samples, as training data for such a system, are difficult to obtain in large quantities, particularly in new platforms. Motivated by this challenge, we aim to design innovative methods for training multimodal CRS effectively even in a small data setting. Specifically, assuming the availability of a small number of samples with dialogue states, we devise an effective dialogue state encoder to bridge the semantic gap between conversation and product representations for recommendation. To reduce the cost of dialogue state annotation, a semi-supervised learning method is developed to effectively train the dialogue state encoder with a small set of labeled conversations. In addition, we design a correlation regularisation that leverages knowledge in the multimodal product database to better align textual and visual modalities. Experiments on the dataset MMD demonstrate the effectiveness of our method. Particularly, with only 5% of the MMD training set, our method (namely SeMANTIC) obtains better NDCG scores than those of baseline models trained on the full MMD training set.

## CCS CONCEPTS

• **Computing methodologies → Discourse, dialogue and pragmatics**; • **Information systems → Recommender systems**; **Multimedia and multimodal retrieval**.

## KEYWORDS

conversational recommendation systems, dialogue states, semi-supervised learning

## 1 INTRODUCTION

Recently, there has been a growing interest in conversational recommendation systems (CRS). These systems bring together the user-friendly nature of conversational AI and the business potential of recommendation systems, potentially revolutionizing how customers engage with e-commerce platforms. Unfortunately, conventional text-based dialogue systems have inherent limitations in capturing user preferences. In many practical situations, a blend of textual and visual cues allows agents to recommend products

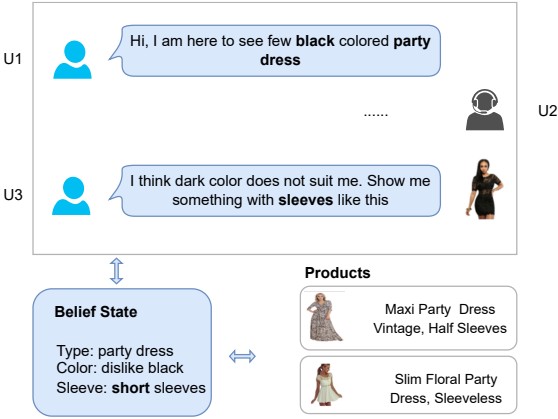

**Figure 1: Multimodal CRS can recommend suitable products based on a user's query, including their preferred example image. The dialogue state (belief state) encapsulates user interests across turns and modalities.**

that are better aligned with user interests. Therefore, multimodal conversational recommendation systems (multimodal CRS) have been introduced (e.g., see Figure 1 for an example).

The advance in deep learning along with the introduction of multimodal benchmarks, such as MMD [27], have contributed significantly to the recent progress in multimodal CRS. A number of methods have been developed using Recurrent Neural Network (RNN) [27], RNN with attention [4], Graph Neural Networks [39], Memory Network [24], Transformer [22], and Graph Attention Network [7]. Unfortunately, deep learning-based methods require a significant number of conversation samples with relevance annotations (for recommendation), which can be challenging to acquire. For example, the aforementioned methods have been trained on MMD using hundreds of thousands of conversations, and it is unclear whether these approaches remain effective when being trained with a smaller sample size.

In this paper, we examine multimodal CRS in a small data setting. Specifically, we consider that there is only a limited number of conversation samples and strive to make the most of the data by following two insights. Firstly, when the number of conversation samples is limited, augmenting them with dialogue states can help align the representations of dialogues and products for better matching. This is supported by the fact that dialogue state tracking (DST) is essential for traditional text-based task-oriented dialogue (TOD) systems [11, 17, 37, 40]. Unfortunately, annotating dialogue states can be time-consuming, particularly in multimodal dialogues. Therefore, we assume that only a subset of conversation samples is annotated with dialogue states and design an effective method for dialogue state modeling. Secondly, the vast amount of products with both textual and visual information should be exploited to

bridge the cross-modal semantic gap. Intuitively, doing so can help improve the system's capability in understanding user preferences across modalities (see U3, Figure 1).

With such considerations, we propose a Sample Efficient multi-modal coNversaTIonal reCommendation system, or SeMANTIC for short. More specifically, dialogue contexts and candidate products are first encoded with a context encoder and a product encoder separately, resulting in initial context/product representations. Such representations are then enhanced with Dialogue State Interaction modules that capture the interactions of the context (or the product) representations with shared dialogue state embeddings, resulting in state-aware representations. By doing so, we leverage dialogue states to align the representations of the dialogue side and the product side. Here, dialogue state embeddings are learned via a teacher-student framework, where the teacher network has access to the limited size of dialogues with ground-truth belief states, and the student network learns from the teacher network to estimate dialogue state embeddings from conversations without dialogue states. We then propose a regularization term that makes state-aware (text/visual) representations of the same product closer to each other. As a result, we effectively utilize the large number of products in the domain database for bridging the cross-modal semantic gap.

All in all, our main contributions are as follows:

- We propose a novel model, SeMANTIC, that enhances dialogue and product representations with dialogue states, and a regularization term that leverages the multimodal product database to bridge cross-modal semantic gap.
- A semi-supervised learning approach is proposed, utilizing the teacher-student framework, to reduce the cost of annotating dialogue states.
- Extensive evaluation on MMD dataset demonstrates the superiority of our model in comparison to strong baselines in a small data setting.
- Further analysis validates that our semi-supervised learning approach is data efficient as it only requires a small ratio of supervision for learning dialogue state embeddings.

## 2 RELATED WORK

### 2.1 MultiModal Conversational Systems

There have been a growing number of studies on multimodal conversational systems thanks to the introduction of multimodal datasets such as SURE [21], FashionIQ [33, 38], MMD [27] and SIMMC [15]. Most of previous methods aim to enhance dialogue representation using different network architectures [22, 25, 27, 39], external knowledge or side information [4, 25, 39], mutual-information [41], knowledge distillation [13], cross-modal interaction or attention [4, 22].

Unlike these studies, we target an under-explored problem of learning effective representations with a limited number of conversations. It is noted that our focus is on grounding dialogues on external data (the recommendation task), which remains challenge particularly now that response generation can be greatly improved with large language models. As dialogue systems are complicated,

it is common for researchers to focus on substaks such as recommendation [24, 39], dense retrieval [31, 34], dialogue State Tracking (DST) [2, 16] for deeper analysis.

### 2.2 Learning in a Small Data Setting

Deep learning has been the mainstream approach recently. Unfortunately, deep learning methods are also data hungry, requiring a large amount of training conversational samples with annotation. For example, to train a conversational recommendation system, it is needed to collect diverse dialogue samples annotated with recommendations and various user requests [1, 19, 20]. As labeled data is difficult to obtain, it is desirable to develop data efficient methods based on pretrained models [10, 36], meta-learning [5], or semi-supervised learning [12, 18, 35].

Our work falls into the semi-supervised learning category but focuses on multimodal dialogues. To the best of our knowledge, our work is the first attempt at this important problem. It should be noted that we cannot simply adopt a unimodal method to a multimodal scenario. For instance, one simple way to apply these available methods [12, 40] to our task is to consider DST as a text sequence generation task. However, as we empirically show in Section 4.5, without careful consideration of the semantic gap between modalities as well as between products and dialogues, even ground-truth (sequentialized) dialogue state will not facilitate the recommendation task.

## 3 METHODOLOGY

*Problem Formalization.* Let $\mathcal{D}_F$ be the set of fully labeled dialogues. Given a dialogue $\tau = \{u_t | 1 \leq t \leq n_\tau\}$, $u_t$ indicates the t-th turn from either the user or the agent. Each utterance $u_t$ contains the textual part $u_t^T$ and the visual part $u_t^I$ (*i.e.* a list of user uploaded images or system recommended product images). For each user turn, utterance is provided with a dialogue state $s_t$ that summarizes the user requests throughout the conversation. Additionally, let $\mathcal{D}_P$ denote the set of partially labeled dialogues for which dialogue state annotations are unavailable. We assume that $\mathcal{D}_P$ is larger in size compared to $\mathcal{D}_F$, but still in a moderate size. The multimodal conversational recommendation task is formalized as selecting appropriate products from a product database $\mathcal{P} = \{(\rho_k^T, \rho_k^I) | 1 \leq k \leq n_\mathcal{P}\}$ as response to a user request. Here, a product in $\mathcal{P}$ is associated with both textual description $\rho_k^T$ and image $\rho_k^I$.

The overall architecture of SeMANTIC is depicted in Figure 2, where the main idea is to treat dialogue states as shared (continuous) variables that bridge the semantic gaps between the textual modality and the visual modality, and between the conversation and the product sides. Specifically, representations of dialogue texts/images and product texts/images are encoded separately by context encoders and product encoders, as discussed in Section 3.1, and then enhanced with dialogue state embeddings using dialogue State Interaction (DSI) modules 3.2. Here, the dialogue state embeddings are obtained by encoding the ground-truth dialogue states for the dialogues in $\mathcal{D}_F$, while they are inferred by the dialogue learner for those in the partially labeled set $\mathcal{D}_P$. Further details about this part are provided in Section 3.5. To address the limitation stemming from the small size of $\mathcal{D}_F$, we introduce a regularization term to

leverage the wealth of information contained within the extensive product database $\mathcal{P}$.

## 3.1 Context and Product Encoders

*Context Encoder.* Let $\tau$ be a dialogue context consisting of $n_\tau$ turns, and $u_t^T = \{w_1, w_2, \ldots, w_{n_t^T}\}$ be the textual utterance at the t-th turn, where $w_i$ is an one-hot representation of the i-th word, we obtain the turn-level text representation as follows:

$$U_{ti}^T = w_i W_{emb} + PE(i)$$
$$U_t^T = \{U_{t1}^T, \ldots, U_{tn_t^T}^T\}$$
$$\hat{U}_t^T = SumPool[SelfAttn(U_t^T, U_t^T, U_t^T)]$$

where $\hat{U}_t^T$ denotes the representation of the textual utterance at the t-th turn, $W_{emb}$ is the pre-trained word embeddings obtained from BERT [6], $PE(\cdot)$ and $SelfAttn(\cdot)$ denote the position embeddings and self-attention [30]. $SumPool[\cdot]$ indicates the sum pooling operation. The dialogue-level representations for the textual modality can be obtained as follows:

$$C^T = \{\hat{U}_1^T, \ldots, \hat{U}_{n_\tau}^T\}$$
$$\hat{C}^T = SelfAttn(C^T, C^T, C^T)$$

Here, $\hat{C}^T = \{\hat{c}_1^T, \ldots, \hat{c}_{n_\tau}^T\}$. Similarly, we construct the turn-level visual representation from the t-th turn $u_t^I = \{v_1, v_2, \ldots, v_{n_t^I}\}$:

$$U_{ti}^I = Linear(ResNet(v_i))$$
$$U_t^I = \{U_{t1}^I, \ldots, U_{tn_t^I}^I\}$$
$$\hat{U}_t^I = SumPool[SelfAttn(U_t^I, U_t^I, U_t^I)]$$
$$C^I = \{\hat{U}_1^I, \ldots, \hat{U}_{n_\tau}^I\}$$
$$\hat{C}^I = CrossAttn(\hat{C}^T, C^I, C^I)$$

$ResNet(\cdot)$ denotes Residual Neural Network [9], and a linear layer $Linear(\cdot)$ is used to project the dimension of $ResNet(v_i)$ from $D^{ResNet}$ to $D^{out}$. Then, the final dialogue-level representations $\hat{c}^T$ and $\hat{c}^I$ (for the textual and visual modalities) are attained from the last turn representations in $\hat{C}^T$ and $\hat{C}^I$ (the $n_\tau$-th representation in $\hat{C}^T$ and $\hat{C}^I$).

*Product Encoder.* The textual $\hat{\rho}^T$ and visual $\hat{\rho}^I$ representations for a product $\rho = (\rho^T, \rho^I)$ are obtained similarly to the turn-level dialogue representations (*i.e.* $\hat{U}_t^T$ and $\hat{U}_t^I$). Note that the $ResNet$ is shared between the context encoder and the product encoder.

## 3.2 Dialogue State Interaction Module

Our objective is to exploit dialogue states to align representations in multimodal CRS. As such, we first get dialogue state embeddings $\tilde{S} \in \mathbb{R}^{n_{state} \times D^{out}}$ from the dialogue context (see Section 3.5 for more details). Inspired by Memory Networks [29], we then introduce Dialogue State Interaction (DSI) modules to enhance both dialogue representations ($\hat{c}^T$ and $\hat{c}^I$) and product representations ($\hat{\rho}^T$ and $\hat{\rho}^I$) with information in dialogue state embeddings.

The general architecture of a DSI module is depicted in Figure 2 with $K$ layers of multi-hop interactions. Given an input vector

$x_k$ and state embeddings $S_k$, the outputs of the k-th layer can be obtained as follows:

$$S_{k+1} = W_{k+1} S_k$$
$$a_{k,i} = \frac{\exp(cos(x_k, S_{k,i}))}{\sum_j^{n_{state}} \exp(cos(x_k, S_{k,j}))}$$
$$x_{k+1} = x_k + \sum_i^{n_{state}} a_{k,i} S_{k+1,i}$$

Here, $W_{k+1}$ and $cos(\cdot)$ denotes the model parameters and cosine similarity, respectively, and $a_k$ corresponds to the softmax attention score vector. Note that $S_0 = \tilde{S}$, and $x_0$ can be either textual or visual representation from a context or product encoder (*i.e.* $\hat{c}^T$, $\hat{c}^I$, $\hat{\rho}^T$ or $\hat{\rho}^I$). As dialogue state embeddings ($\tilde{S}$) are shared for the dialogue context and the product candidate (see Figure 2), DSI module helps align the corresponding representations for effective matching. To be mentioned, we denotes the enhanced dialogue representations from the last layer of DSI modules as $x^{CT}$ and $x^{CI}$, and enhanced product representations as $x^{PT}$ and $x^{PI}$.

## 3.3 Recommendation

Given a dialogue $\tau$ and a candidate product $\rho$, the relevance score is measured using a $tanh(\cdot)$ activation function as follows:

$$f(\tau, \rho) = \tanh(cos(x^{CT}, x^{PT}) + cos(x^{CI}, x^{PI}))$$

As mentioned in Section 3.2, $x^{CT}$, $x^{CI}$, $x^{PT}$ and $x^{PI}$ are enhanced representations of $\hat{c}^T$, $\hat{c}^I$, $\hat{\rho}^T$ and $\hat{\rho}^I$.

## 3.4 Training

To train SeMANTIC, we construct a training set by sampling dialogue contexts and the gold image responses from $\mathcal{D}_P$. Given a training sample $\{(\tau, \rho_1^+, \ldots, \rho_{n_{pos}}^+, \rho_1^-, \ldots, \rho_{n_{neg}}^-)\}$, $\tau$ indicates one conversation context, whereas $\rho_j^+$ and $\rho_j^-$ denote a positive recommendation and a (sampled) negative recommendation for $\tau$. Note also that the dialogue state encoder is trained jointly with the rest of the model. However, we postpone the detailed discussion until Section 3.5, where semi-supervised learning for dialogue state modeling is described.

*Ranking Loss.* The main objective for training SeMANTIC is to maximize the margin in the relevance score $f(\tau, \rho)$ of the positive product compared to the negative product. In other words, we minimize the following ranking loss:

$$\mathcal{L}_{rank} = max(0, 1 - f(\tau, \rho^+) + f(\tau, \rho^-))$$

where the loss is measured for a sample triple $(\tau, \rho^+, \rho^-)$. Here, we drop the subscripts of products for simplicity.

*Jensen Shannon Divergence.* To better align the context and the product representations, we measure Jensen-Shannon divergence [23] between the attention vectors extracted from the $K + 1$ layer of DSI Modules ($a_k$ in Section 3.2 for $k = K+1$). Specifically, we respectively obtain ($a^{CT}, a^{CI}$) for the enhanced dialogue representations ($x^{CT}, x^{CI}$), and $a^{PT}, a^{PI}$) for the enhanced product representations ($x^{PT}, x^{PI}$), then measure:

$$g(\tau, \rho) = JS(a^{CT}, a^{PT}) + JS(a^{PI}, a^{PI})$$

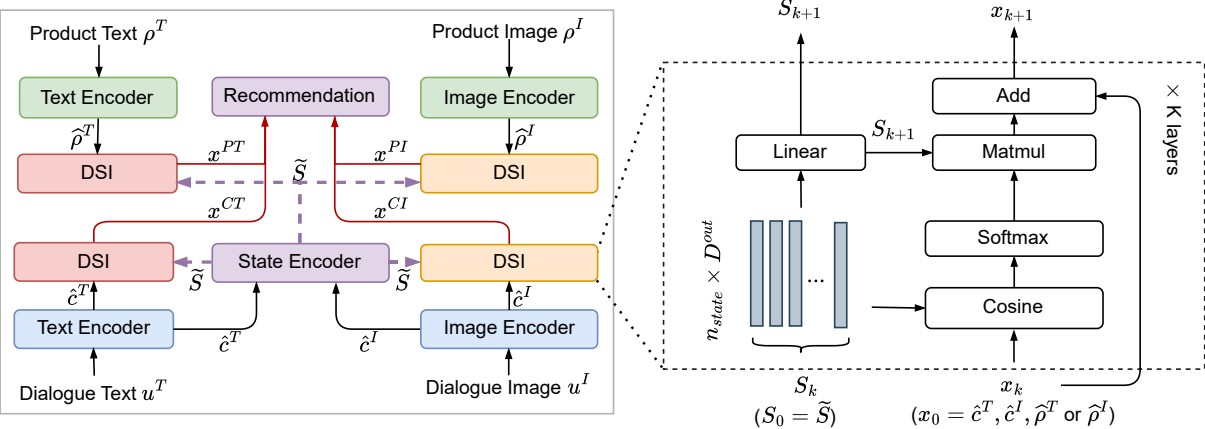

**Figure 2: The overall architecture of SeMANTIC is illustrated on the left. Initially, dialogue text/image and product text/image are separately encoded through Text Encoders and Image Encoders, respectively. Subsequently, a State Encoder is applied to encode dialogue state embeddings $\widetilde{S}$. Following this, Dialogue State Interaction (DSI) modules are employed to enhance the embeddings of dialogue ($\hat{c}^T$ and $\hat{c}^I$) and product ($\hat{\rho}^T$ and $\hat{\rho}^I$) with $\widetilde{S}$, resulting in the final representations ($x^{CT}$, $x^{CI}$, $x^{PT}$ and $x^{PI}$) for making recommendations (DSI modules of the same color are shared between the dialogue side and product side). Further details of a DSI module are provided in the right block, where "Cosine" and "Matmul" refers to cosine similarity and matrix multiplication.**

Intuitively, we would like the $g$ score to be small for the relevant pair $(\tau, \rho^+)$ and larger for the irrelevant pair $(\tau, \rho^-)$. To achieve this, we incorporate the following loss to the objective function:

$$\mathcal{L}_{JS} = max(0, g(\tau, \rho^+) - g(\tau, \rho^-))$$

*Correlation Similarity.* Due to the limited size of conversational samples, we rely on the larger number of available products to bridge the gap between the textual and visual modalities. Our goal is to minimize the regularization term calculated for a given product $\rho$ as follows:

$$\mathcal{L}_{co\_sim}(\rho) = max(0, 1 - cos(x^{PT}, x^{PI}))$$

The idea here is make the (text/visual) state-enhanced representations of the same product closer to each other.

*Overall.* Finally, the overall loss function $\mathcal{L}_{all}$ is:

$$\mathcal{L}_{all} = \mathcal{L}_{rk} + \mathcal{L}_{JS} + \sum_{\rho_i^{\pm}} \mathcal{L}_{co\_sim}(\rho_i^{\pm})$$

where $\rho_i^{\pm}$ indicates either a positive or negative sample associated with the context $\tau_i$.

### 3.5 Semi-supervised State Learning

To leverage small samples with dialogue states, we adopt the teacher-student framework [3], where both the teacher and student possess similar structures (as depicted in Figure 2) but differ in the State Encoder (as illustrated in Figure 3).

*Teacher Network.* The teacher has access to the ground truth dialogue state in $\mathcal{D}_F$, where each dialogue state $u^S = [(u_i^{SK}, u_i^{SV}) | 1 \le i \le n_{state}]$ is a list of slot and value pairs. The slot keys are drawn from a predefined set of $n_{state}$ product properties defined in the product database $\mathcal{P}$, such as color or type. For each slot key such as

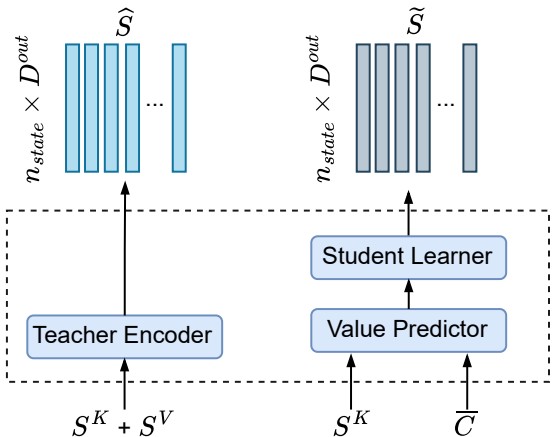

**Figure 3: The architecture of the State Encoder, where the teacher and student networks have different structures. The teacher network is depicted on the left (Teacher Encoder), and the student network is shown on the right (Value Predictor and Student Learner). Since the dialogue state comprises a list of slot key and value pairs, $S^K$ and $S^V$ represent the embeddings of keys and values, respectively. $\bar{C}$ denotes the summation of $\hat{C}^T$ and $\hat{C}^I$ which represents the information of the entire context. $\hat{S}$ refers to the ground-truth state embeddings, while $\hat{S}$ represents the predicted state embeddings.**

color, the slot value is "none" if it is not mentioned in the dialogue context $\tau$, and a specific value (e.g. red) otherwise. For the i-th slot, we treat the slot key and value as strings and attain the key embeddings $S_i^K \in \mathbb{R}^{1 \times D^{out}}$ and value embeddings $S_i^V \in \mathbb{R}^{1 \times D^{out}}$ via pre-trained word embeddings in BERT and pooling, which is

similar to the text encoder in Section 3.1. The state embeddings are then obtained via $SelfAttn(\cdot)$ as follows:

$$S_i = S_i^K + S_i^V$$
$$S = \{S_1, \ldots, S_{n_{state}}\}$$
$$\hat{S} = SelfAttn(S, S, S)$$

*Student Network.* The student network estimates the slot value embedding from the context information by employing a "Value Predictor". Specifically, We first obtain the key embeddings $S_i^K$ for all slot keys, following a similar approach to that in the teacher network. The state value embeddings are then calculated as follows:

$$\bar{C} = \hat{C}^T + \hat{C}^I$$
$$\bar{S}^V = CrossAttn(S^K, \bar{C}, \bar{C})$$

Here, $CrossAttn(\cdot)$ represents the cross-attention operator. We subsequently derive the predicted dialogue state embeddings $\widetilde{S}$ using the "Student Learner" as follows:

$$\bar{S} = S^K + \bar{S}^V$$
$$\widetilde{S} = SelfAttn(\bar{S}, \bar{S}, \bar{S})$$

*Joint Training.* We train the teacher network on $\mathcal{D}_F$ and the student network on $\mathcal{D}_F + \mathcal{D}_P$ using the loss $\mathcal{L}_{all}$ as in Section 3.4. Hereafter, we refer to the teacher and the student training losses as $\mathcal{L}_{all}^{tea}$ and $\mathcal{L}_{all}^{stu}$. We then let the teacher guide the student network by minimizing the mean square error $\mathcal{L}_{MSE}$ measured between ground-truth dialogue state embeddings $\hat{S}$ and the predicted state embeddings $\widetilde{S}$ on $\mathcal{D}_F$. The joint training objective, therefore, is:

$$\alpha \mathcal{L}_{all}^{tea} + (1-\alpha) \left\{ \mathcal{L}_{all}^{stu} + \sum_{\tau \in \mathcal{D}_F} MSE(\hat{S}, \widetilde{S}) \right\}$$

where $\hat{S}$ and $\widetilde{S}$ represent the outputs of the teacher and student networks, respectively.

# 4 EXPERIMENTS

## 4.1 Dataset

**Table 1: Statistics of the dataset by [25] (MMD-v2) and the subset with dialogue state annotation (MMD-v3 with DS). "Avg Rec Turns" indicates the average number of recommendation turns in each dialogue. "Avg Pos Imgs" and "Avg Neg Imgs" represent average number of $\rho^+$ and $\rho^-$ in Section 3.4, respectively.**

| Dataset | MMD-v2 | | | MMD-v3 with DS | | |
|---|---|---|---|---|---|---|
| Dataset Stats | Train | Valid | Test | Train | Valid | Test |
| dialogues | 105439 | 22595 | 22595 | 5478 | 1113 | 1174 |
| Proportion | 70% | 15% | 15% | 72% | 14% | 14% |
| Avg Rec Turns | 5 | 5 | 5 | 6 | 6 | 6 |
| Avg Pos Imgs | 4 | 4 | 4 | 4 | 4 | 4 |
| Avg Neg Imgs | 616 | 618 | 994 | 628 | 632 | 989 |

Experiments are conducted on MMD [27]. The MMD dataset contains more than 150k conversations in retail domain. Following previous works [24, 39], we adopt the updated MMD dataset constructed by Nie [24] and refer to it as MMD-v2, which is divided into training/validation/test sets with ratio 70%/15%/15%. To study the impact of the sample size and dialogue states, we select around 7765 samples (5% of MMD-v2) and perform dialogue state annotation with slot keys being product attributes. We refer to this set of MMD as MMD-v3. We split the data to sets train/valid/test so that the training/valid/test set of MMD-v3 is a subset of the corresponding set of MMD-v2. More details are presented in the Table 1.

We conducted additional experiment on SIMMC, a dataset with the size similar to that of MMD-v3, and obtained similar observations with those on MMD-v3. Therefore, we put the experimental results on SIMMC to the supplementary document.

## 4.2 Experimental Settings

*Implementation Details.* We implement our proposed model using PyTorch[1] and conduct our experiments on 1 NVIDIA V100 GPU with a mini-batch size 64 and 50 epochs. Adam [14] is adopted as the optimizer, with the initial learning rate $5 \times 10^{-4}$ and the linear learning rate scheduler [8] is used. The dimension of the initial word embedding is set to 768, and the dimension of the initial image embedding $D^{ResNet}$ is 512. The dimension of both context representation and product representation $D^{out}$ is set to 768. The number of layers of all transformer based encoders and decoders are set to 3, the number of attention heads in the multi-head attention is 8 and the inner-layer size is 768. We set all dropout rate to 0.1 [28], and $\alpha$ to 0.5 (Section 3.5). Moreover, we use 5 turns prior to the current turn as the context with the maximum sentence length of 30 and the maximum number of historical images to 5.

Following [39], we set the ratio of positive to negative products to 1:4 and 5:1000 for training and testing, respectively. Results of all experiments evaluated on MMD-v3 are averaged over fivefold cross validation. For experiments where we train on MMD-v3 and test on MMD-v2, we split the train/valid/test so that the training/valid/set set of MMD-v3 is a subset of the training/valid/testing set of MMD-v2. In such experiments, multiple runs of SeMANTIC and the baselines are from different random initialization seeds.

It is worth mentioning that although both $\mathcal{L}_{all}^{teacher}$ and $\mathcal{L}_{all}^{student}$ contain $\mathcal{L}_{JS}$ and $\mathcal{L}_{co\_sim}$, such losses are calculated by the teacher model and deactivated by the student model on $\mathcal{D}_F$. These losses are only activated for the student model on $\mathcal{D}_P$.

*Evaluation Metrics.* Following [24, 39], Precision@k, Recall@k, and NDCG@k for (k=5, 10, and 20) are the adopted metrics for the recommendation task in multimodal CRS.

*Compared Methods.* We compare our method SeMANTIC to baselines with published codes. For CLIP, we only fine-tune its final linear projector and add self-attention layers to encoder turn-level text embeddings and image embeddings. Then we concatenate text embeddings and image embeddings as the final context embeddings and product embeddings. For other baseline methods, we adhere to a standardized approach which adopts the default configurations as set in the original papers. By doing so, we ensure a consistent and accurate comparison with the established methodology.

- **MHRED**: Saha et al. [27] present a basic multimodal hierarchical encoder-decoder model as the first benchmark in the field of multimodal CRS.

---

[1]https://pytorch.org/

**Table 2: The overall results of SeMANTIC and baselines, in which the average and standard deviations of different folds are reported. MMD-v3/v2 (or MMD-v3/v3) means we train the model on the training set of MMD-v3 and evaluate on the testing set of MMD-v2 (or MMD-v3). TREASURE† and Enteract† are both trained and tested on MMD-v2 and reported from [39] and [7]. Our method outperforms other baselines trained on MMD-v3 and achieves a higher NDCG@20 score than TREASURE trained on MMD-v2.**

| | Method | Precision@5 | Recall@5 | NDCG@5 | Precision@10 | Recall@10 | NDCG@10 | Precision@20 | Recall@20 | NDCG@20 |
|---|---|---|---|---|---|---|---|---|---|---|
| MMD-v3/v3 | MHRED | 34.56±1.50 | 40.91±1.83 | 39.09±1.35 | 20.54±0.79 | 48.55±1.92 | 42.60±1.33 | 12.14±0.42 | 57.35±1.94 | 45.82±1.31 |
| | UMD | 27.13±4.80 | 30.04±4.71 | 25.62±4.08 | 18.13±2.06 | 42.52±4.61 | 31.23±3.87 | 11.82±0.81 | 55.27±3.67 | 35.89±3.42 |
| | MAGIC | 46.33±0.77 | 53.48±0.94 | 51.61±1.87 | 26.21±0.34 | 60.72±0.83 | 54.86±1.55 | 14.39±0.19 | 66.93±0.93 | 57.10±1.44 |
| | CLIP | 14.10±0.19 | 16.96±0.33 | 16.81±0.37 | 8.71±0.12 | 20.88±0.43 | 18.63±0.41 | 5.47±0.08 | 26.11±0.52 | 20.60±0.43 |
| | LARCH | 30.64±2.57 | 37.00±2.93 | 36.66±3.25 | 21.22±1.23 | 50.23±2.77 | 43.56±2.94 | 13.01±0.36 | 61.25±1.59 | 48.00±2.53 |
| | TREASURE | 45.75±1.47 | 53.34±1.78 | 52.11±2.10 | 25.59±0.55 | 59.82±1.31 | 55.36±1.95 | 14.15±0.19 | 66.37±0.91 | 57.46±1.73 |
| | Enteract | 49.59±0.62 | 55.30±0.54 | 47.41±0.78 | 29.93±0.17 | 65.74±0.25 | 50.99±0.61 | 17.03±0.07 | 74.92±0.24 | 53.53±0.60 |
| | SeMANTIC | **63.87**±0.39 | **75.19**±0.54 | **75.87**±0.71 | **32.96**±0.16 | **77.71**±0.53 | **76.94**±0.72 | **17.06**±0.09 | **80.52**±0.47 | **77.91**±0.71 |
| MMD-v3/v2 | MHRED | 30.66±3.00 | 35.30±3.71 | 36.47±3.31 | 18.51±1.43 | 44.08±3.36 | 39.87±3.22 | 10.97±0.64 | 52.29±3.08 | 42.85±3.09 |
| | UMD | 13.49±0.66 | 15.66±1.59 | 15.00±1.81 | 10.74±0.22 | 24.93±1.39 | 18.68±1.55 | 7.81±0.76 | 35.97±2.72 | 22.76±1.68 |
| | MAGIC | 38.31±1.77 | 44.88±2.06 | 43.38±2.60 | 22.08±0.62 | 51.86±1.44 | 46.46±2.34 | 12.48±0.22 | 58.85±1.02 | 48.96±2.16 |
| | CLIP | 12.08±0.32 | 14.82±0.29 | 15.39±0.33 | 7.22±0.19 | 17.64±0.31 | 14.37±4.89 | 4.49±0.11 | 21.81±0.37 | 18.24±0.37 |
| | LARCH | 23.61±1.42 | 28.55±1.66 | 29.39±1.95 | 16.90±0.52 | 40.02±1.16 | 35.32±1.71 | 10.71±0.12 | 50.41±0.56 | 39.51±1.44 |
| | TREASURE | 34.99±1.74 | 41.06±2.05 | 39.75±1.79 | 20.47±0.72 | 48.04±1.81 | 42.88±1.65 | 11.85±0.36 | 55.73±1.85 | 45.66±1.62 |
| | Enteract | 41.65±0.96 | 46.49±1.19 | 41.00±1.71 | 24.59±0.52 | 54.13±1.40 | 43.67±1.79 | 15.19±0.22 | 65.88±1.28 | 46.88±1.72 |
| | SeMANTIC | **58.66**±0.32 | **69.66**±0.34 | **71.08**±0.65 | **30.29**±0.09 | **72.06**±0.17 | **72.08**±0.59 | **15.66**±0.06 | **74.60**±0.24 | **72.94**±0.59 |
| | TREASURE † | 59.87 | 71.39 | 71.24 | 31.34 | 74.85 | 72.72 | 16.33 | 78.17 | 72.87 |
| | Enteract † | 61.69 | 67.79 | 58.81 | 32.44 | 71.26 | 60.05 | 16.72 | 73.68 | 60.79 |

- **UMD**: Cui et al. [4] first propose a user attention-guided multimodal CRS which is based on MHRED and uses a hierarchical product taxonomy tree to extract visual features.
- **MAGIC**: Nie et al. [25] propose knowledge-aware RNN to encode dialogue context for response generation and product recommendation task.
- **LARCH** Nie et al. [24] utilize a multimodal hierarchical graph-based neural network to better characterize dialog context representation. Additionally, LARCH exploits a multiform knowledge embedding memory network to unify heterogeneous knowledge (*i.e.* style-tips, product attributes) into a homogeneous base, and enhances dialog contexts with such information.
- **TREASURE** Zhang et al. [39] utilize graph attention network to represent the context of the dialogue, where each turn is deemed as a node within the graph. Additionally, TREASURE encodes a textual sequence using an attribute-enhanced textual encoder, allowing the model to focus on attribute-related keywords.
- **Enteract** Du et al. [7] focus on product representations based on a gated multi-view image encoder and graph attention network. They also model two forms of inter-modal interactions for product representations.
- **CLIP** Radford et al. [26] present a powerful pre-trained multimodal model for image-text retrieval. We fine-tune its final linear projector and add self-attention layers to encoder turn-level text embeddings and image embeddings. We then concatenate text embeddings and image embeddings as the final context embeddings and product embeddings.

Except for CLIP, other baselines have been tested on MMD datasets. As such, we adopt the default configurations as set in the original papers of such baselines for fair comparison.

*Experimental Design.* Our experiments are designed to answer the following research questions: 1) **RQ1**: How do SeMANTIC and other baselines perform when being trained with small conversational sample sets? (Section 4.3); 2) **RQ2**: How is the effectiveness of SeMANTIC when only smaller samples are labeled with dialogue states? (Section 4.4); 3) **RQ3**: Do baselines effectively exploit dialogue states if we provide them with grouth-truth dialogue states during testing? (Section 4.5).

## 4.3 Main Results

We consider the case when the number of conversational samples is in the scale of MMD-v3, which is much smaller compared to MMD-v2. Note that on MMD, all compared models are trained on MMD-v3 but tested on MMD-v3 or MMD-v2. In addition, we consider $\mathcal{D}_F = \mathcal{D}_P$ = MMD-v3 for SeMANTIC here, leaving the analysis for different ratios of these two sets to next section.

Table 2 presents the experimental results averaged over five-fold cross-validations, where a number of observations can be drawn. Firstly, SeMANTIC outperforms the compared methods on two testing sets of MMD, partially validating its effectiveness and generalization. Secondly, even though we train our method with MMD-v3, which is only 5% of the training set of TREASURE† (trained on MMD-v2), the evaluation results on the test set of MMD-v2 show that our method is comparable to TREASURE† on NDCG@5, NDCG@10 , and even better on NDCG@20. It is important to note that training on MMD-v2 is time-consuming, and we are unable to replicate it due to the invalidation of image URLs over time (Du et al. [7] have filtered out invalid images and resampled the MMD-v2 dataset to train Enteract), which prevents us from training comparable models multiple times for comparison. As a result, we directly report the results of TREASURE † from [39]. Last but not least, although CLIP is a robust pre-trained model for image-text

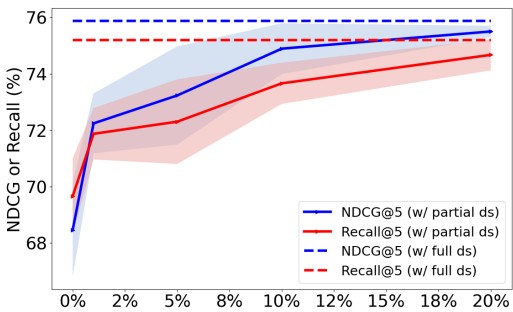

**Figure 4: Performance of SeMANTIC trained with varying sizes of fully labeled data on MMD-v3. "w/ full ds" indicates that $\mathcal{D}_F=\mathcal{D}_P=$MMD-v3**

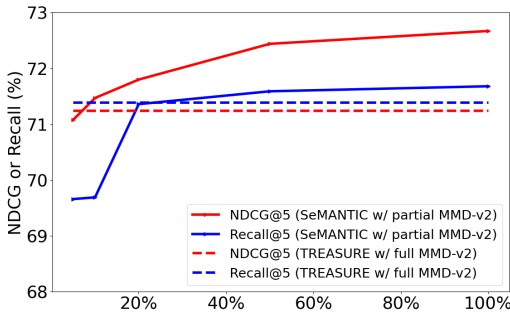

**Figure 5: Performance of SeMANTIC trained with varying sample sizes on MMD-v2. For SeMANTIC, $\mathcal{D}_F$ is MMD-v3, and $\mathcal{D}_P$ corresponds to a different percentage of MMD-v2. In contrast, TREASURE is trained on entire MMD-v2 dataset. Both SeMANTIC and TREASURE are tested on MMD-v2.**

retrieval, it does not perform well in our specific task and domain. This shows that research into sample-efficient methodologies are still relevant in the context of large pretrained models as there are many domains without redundant training dataset. In fact, current studies in LLM start focusing on semi-supervised methods to mitigate the issue of limited set of instruction data [32].

### 4.4 The Impacts of Sample Size

To verify the effectiveness of semi-supervised dialogue state learning, we conduct experiments on MMD-v3 ($\mathcal{D}_F=\mathcal{D}_P=$MMD-v3) and change the size of $\mathcal{D}_F$ from 0% to 20% of MMD-v3. For every epoch, we first jointly train both teacher and student models on $\mathcal{D}_F$, then train the student model on $\mathcal{D}_P$ without considering ground-truth dialogue state. Figure 4 indicates that our model improves as more annotated data is utilized. Furthermore, the reduction in standard deviation indicates that the model's performance becomes more stable as more samples with labeled dialogue states are considered. More importantly, our model's performance with 20% of the supervision ratio is nearly as good as having full supervision to learn state embeddings.

We evaluate the impact of the number of training samples by conducting experiments on MMD-v2. Specifically, we keep $\mathcal{D}_F$ to be MMD-v3 training set, and increase the set $\mathcal{D}_P$ to include more samples from the training set of MMD-v2. The results of SeMANTIC

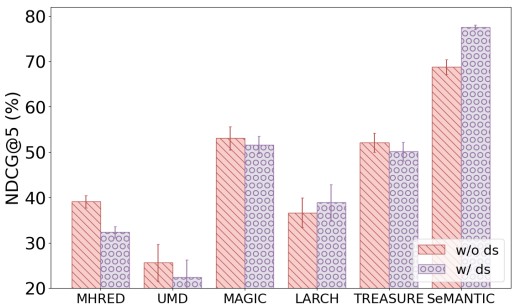

**Figure 6: The impacts of dialogue states on SeMANTIC and compared methods, tested on MMD-v3. We can see that only SeMANTIC and LARCH can benefit from dialogue states (ds).**

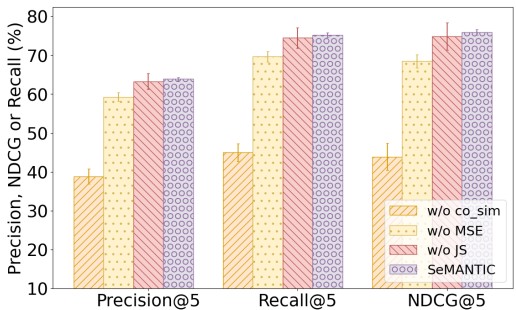

**Figure 7: The impacts of different loss functions on SeMAN-TIC are tested on MMD-v3. "co_sim" indicates $\mathcal{L}_{co\_sim}$, "MSE" indicates $\mathcal{L}_{MSE}$ and "JS" indicates $\mathcal{L}_{JS}$. Since the absence of any loss hurts the overall performance, all loss functions contribute to the recommendation accuracy.**

and TREASURE are then reported on the testing set of MMD-v2 in Figure 5. The results show that SeMANTIC outperforms TREASURE in terms of NDCG@5 when the size of $\mathcal{D}_P$ to be around 10% of the MMD-v2, validating the sample efficiency of SeMANTIC.

### 4.5 Can Baselines Benefit from Dialogue States?

We study whether the incorporation of dialogue states into baselines can help improve performance of such methods. As adapting the baselines to incorporate dialogue state prediction is nontrivial, we directly consider ground truth dialogue states as part of the dialogue input for the baselines during both training and testing. As SeMANTIC (w/ ds) only exploits ground-truth values during training, this setting gives baseline methods considerable advantage. This experiment is trained and tested on MMD-v3. For SeMAN-TIC (w/o ds), state encoding excludes slot values during training, making it fair to compare with the baselines (w/o ds).

The performance comparison between the baselines and Se-MANTIC with and without dialogue states is presented in Figure 6. Among all the methods, only LARCH and SeMANTIC show improvement on NDCG@k (k=5,10) when dialogue states are considered. One possible explanation is that the slot values in dialogue states may not match product attribute values. As a result, only LARCH, which leverages diverse interactions between dialogues

**Table 3: The results of SeMANTIC with different $\alpha$ on MMD-v3. The results show that our method is not sensitive to $\alpha$.**

| Param $\alpha$ | Recall@5 | Recall@10 | Recall@20 |
|:---:|:---:|:---:|:---:|
| $\alpha = 0.1$ | 73.57±1.59 | 74.81±1.64 | 75.85±1.55 |
| $\alpha = 0.3$ | 74.04±1.64 | 75.27±1.69 | 76.22±1.67 |
| $\alpha = 0.5$ | 75.87±0.71 | 76.94±0.72 | 77.91±0.71 |
| $\alpha = 0.7$ | 75.65±1.71 | 76.77±1.79 | 77.74±1.73 |
| $\alpha = 0.9$ | 75.69±0.78 | 76.91±0.61 | 77.84±0.60 |

**Table 4: Human evaluation for SeMANTIC vs TREASURE: the evaluation is measured per recommendation (Rec. Cases). SeMANTIC outperforms TREASURE in 30% of cases.**

| | Win | Tie | Lose |
|:---:|:---:|:---:|:---:|
| **Rec. Cases** | 32.20% | 63.84% | 5.98% |

and knowledge, and SeMANTIC, which incorporates correlation similarity, can make good use of dialogue state information.

### 4.6 Ablation Study

To assess the impacts of different loss functions, we exclude correlation similarity loss $\mathcal{L}_{co\_sim}$ (w/o $co\_sim$), MSE loss $\mathcal{L}_{MSE}$ (w/o $MSE$), or JS divergence $\mathcal{L}_{JS}$ (w/o $JS$) from the training objective.

Figure 7 illustrates the impact of different loss functions on SeMANTIC, as measured on MMD-v3. The results reveal several findings. Firstly, the extraction of hidden information from text-image correlation in products (co_sim) and MSE loss are essential in enhancing the model's performance, as evidenced by the decline in performance when this information is omitted. Secondly, the incorporation of $\mathcal{L}_{JS}$ helps reduce variation, making the performance more stable. This is evident as the exclusion of $\mathcal{L}_{JS}$ (w/o JS) leads to larger error bars in Figure 7.

To study the effect of hyper-parameter $\alpha$, we did several experiments with different $\alpha$ on MMD-v3. The results with different $\alpha$ are given in Table 3, which shows that our method is not sensitive to hyper-parameter $\alpha$.

### 4.7 Human Evaluation and Case Study

To evaluate the effectiveness of our method, we conducted a human evaluation comparing its recommendation results against those of TREASURE [39]. We randomly sampled 60 recommendation turns from the MMD dataset. Three participants were recruited, each presented with recommendation results from both methods without knowledge of the method identities. We then calculated the ratio of cases where SeMANTIC wins/ties/loses to TREASURE across all votes. As shown in Table 4, the results show that SeMANTIC wins in 32% of cases and ties in 63% of cases to TREASURE.

In Figure 8(a), SeMANTIC surpasses TREASURE by providing the highest number of correct images. Additionally, in Figure 8(b), both SeMANTIC and TREASURE accurately select images, but SeMANTIC also places them at the top positions. In Figure 8(c), despite

**Question**: Can you show me a few of your top knit woven pantyhose?

SeMANTIC

TREASURE

(a) Case Win

**Question**: Can you show me some of your blouse having an loop type closure?

SeMANTIC

TREASURE

(b) Case Tie

**Question**: I intend to see gloves having natural shape for myself.

SeMANTIC

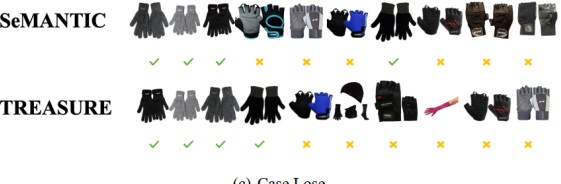

TREASURE

(c) Case Lose

**Figure 8: Top-10 image response selection results of our Se-MANTIC and TREASURE in cases of win, tie, and loss. Images with a check mark indicate the ground-truth recommendations. Even in the case of loss, SeMANTIC ranks ground-truth recommendations better than TREASURE.**

SeMANTIC receiving lower ratings in human evaluation, it consistently prioritizes ground-truth relevant items at the top positions, showcasing the superiority of our method over TREASURE.

## 5 CONCLUSION AND FUTURE WORK

This paper presents a novel approach named SeMANTIC for multimodal CRS. To align multimodal representations, we propose dialogue state interaction modules to enhance both the dialogue and the product sides with dialogue states. To overcome the challenge of collecting dialogue state labels, we develop a teacher-student framework to learn dialogue state embeddings during inference. In addition, we introduce correlation regularization for semantic alignment on the abundant products in the database. Our thorough experiments demonstrate the superiority of our method in the recommendation task when compared to existing methods.

Our method can be adapted to reduce the sample collection cost for general multimodal dialogues. For instance, one can consider dialogue summaries instead of "dialogue states" as the bridge for aligning multimodal dialogue representations. Those enhanced representations can then be used for downstream tasks such as external (textual/visual) knowledge retrieval or response generation.

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
