# OpenReview forum: "Sample Efficiency Matters: Training Multimodal Conversational Recommendation Systems in a Small Data Setting"
_acmmm.org/ACMMM/2024/Conference — MM2024 Poster_

### Official Review · Reviewer_WfUF · 2024-05-23

**Rating:** 4
**Confidence:** 3

**Summary:**

This paper proposed a new method SeMANTIC, which is specially designed for training multimodal conversational recommendation systems with a small amount of data. To achieve this, the model uses semi-supervised training with a teacher-student framework for the dialogue state encoder, and uses the product database for multimodal information alignment. Experiment results show that if all are trained on a small subset of data, the proposed method can achieve better performance than other baselines.

**Strengths:**

-	The proposed method has shown to be effective with small amounts of training data, it can outperform other sota baselines in multimodal CSR, and achieve similar results to those fully trained models.
-	The paper did a comprehensive study on the effectiveness of SeMANTIC regarding training sample size, and the two designed modules. It is notable that the model with a 20% supervision ratio is nearly as good as having full supervision.
-	The paper is overall well-organized and clearly written.

**Limitations:**

-	Can the author further explain why the alignment of textual and visual information can also be achieved so effectively, even given only a small subset of training data?
-	A brief description of hyperparameter optimization (if done), architecture selection (n. of layers), and training time should be provided.

**Suitability:**

3

---

### Official Review · Reviewer_wfgN · 2024-05-24

**Rating:** 3
**Confidence:** 3

**Summary:**

This paper focused on the multimodal conversational recommendation systems (multimodal CRS) under a small data setting.  It introduces SeMANTIC, an approach that enhances dialog and product representations with dialog states and employs a regularization term to bridge the cross-modal semantic gap. To reduce the cost of dialogue state annotation, a semi-supervised learning method is developed to effectively train the dialogue state encoder with a small set of labelled conversations. Experimental results on the MMD dataset showed the effectiveness of the proposed approach.

**Strengths:**

1. The approach obtained impressive results under low-resource setting, which may facilitate the future application of this technique.
2. Experiments and discussions are extensive.

**Limitations:**

1. I notice that in the supplemented material, the paper ignored comparing with the most recent work Enteract [1]. However, from [1], it seems that the performance of Enteract is better than the proposed approach. I would suggest the author to supplement the result and discuss the reason.
- [1] Du et al, Enhancing Product Representation with Multi-form Interactions for Multimodal Conversational Recommendation. MM 2023.

2. Similar, based on Table 2, since the most recent and stronger baseline is Enteract, I would suggest the author should also compare with it in Figure 7, and Figure 8, instead of comparing with the weak baseline TREASURE.

**Suitability:**

2

---

### Official Review · Reviewer_zeGB · 2024-05-27

**Rating:** 4
**Confidence:** 3

**Summary:**

This paper addresses the challenge of training multimodal conversational recommendation systems (CRS) with limited conversational samples. The authors propose a method that includes an effective dialogue state encoder and a semi-supervised learning approach to bridge the semantic gap between conversation and product representations, reducing the need for extensively annotated dialogue states. Their method, SeMANTIC, demonstrates superior performance on the MMD dataset, achieving better NDCG scores with less training data compared to baseline models trained on the full dataset.

**Strengths:**

1. Various baselines are included in comparison with the proposed method SeMANTIC.

2. A comprehensive ablation study is conducted to evaluate different components in SeMANTIC.

3. Several qualitative evaluation examples are included.

**Limitations:**

1. One other line of work about few-shot compositional multimodal learning [1-3] could be highly relevant to SeMANTIC, which can be missing in both the related work section and as comparative baselines.

2. More recently developed MLLMs [4-6] can also be relevant to the proposed task with good zero-shot or few-shot performance. Could the authors also discuss these works?

[1] Wu, Junda, et al. "Few-shot composition learning for image retrieval with prompt tuning." Proceedings of the AAAI Conference on Artificial Intelligence. Vol. 37. No. 4. 2023.

[2] Nayak, Nihal V., Peilin Yu, and Stephen Bach. "Learning to Compose Soft Prompts for Compositional Zero-Shot Learning." The Eleventh International Conference on Learning Representations. 2022.

[3] Lu, Xiaocheng, et al. "Decomposed soft prompt guided fusion enhancing for compositional zero-shot learning." Proceedings of the IEEE/CVF Conference on Computer Vision and Pattern Recognition. 2023.

[4] Alayrac, Jean-Baptiste, et al. "Flamingo: a visual language model for few-shot learning." Advances in neural information processing systems 35 (2022): 23716-23736.

[5] Li, Juncheng, et al. "Fine-tuning multimodal llms to follow zero-shot demonstrative instructions." The Twelfth International Conference on Learning Representations. 2023.

[6] Yin, Zhenfei, et al. "Lamm: Language-assisted multi-modal instruction-tuning dataset, framework, and benchmark." Advances in Neural Information Processing Systems 36 (2024).

**Suitability:**

3

---

### Meta-Review · Area_Chair_UgJq · 2024-07-03

**Recommendation:** Accept (Poster)
**Confidence:** 4

**Metareview:**

This paper proposes a sampling strategy to improve the training of multimodal conversational recommendation systems, in data-limited regimes.

The paper got borderline positive reviews, with three Borderline Accepts.

Among its strengths, reviewers highlight:
- The proposed SeMANTIC demonstrated to be effective in a low-resource setting (wfgN, WfUF) - which was the primary goal with which SeMANTIC was designed - and was compared with several baselines (zeGB).
- The experiments are extensive (zeGB, wfgN, WfUF).
- The paper is well-organized and clearly written (WfUF).

As for its weaknesses:
- Relevant work is overlooked and left out from the established comparisons (zeGB, wfgN)
- Authors should have compared against Enteract in Figures 7 and 8 (wfgN). This concern was addressed in the rebuttal.

Overall, the paper has its merits, with reviewers recognizing the extensive evaluation and effectiveness of the proposed approach.
Provided that authors include the clarifications discussed in their rebuttal in the final manuscript (in particular w.r.t. the related work pointed out by reviewers and comparison with the stronger baseline, Enteract), I suggest this work be accepted as Poster.